# Functionalization with Polyphenols of a Nano-Textured Ti Surface through a High–Amino Acid Medium: A Chemical–Physical and Biological Characterization

**DOI:** 10.3390/nano12172916

**Published:** 2022-08-24

**Authors:** Rafaella C. P. Scannavino, Giacomo Riccucci, Sara Ferraris, Gabriel L. C. Duarte, Paulo T. de Oliveira, Silvia Spriano

**Affiliations:** 1School of Dentistry of Ribeirão Preto, University of São Paulo, Ribeirão Preto 14040-904, SP, Brazil; 2Department of Applied Science and Technology, Politecnico di Torino, 10126 Torino, Italy

**Keywords:** titanium, functionalization, polyphenols, implants

## Abstract

The study aimed to identify an effective mechanism of adsorption of polyphenols on a nano-textured Ti surface and to evaluate the osteogenic differentiation on it. The source of polyphenols was a natural extract from red grape pomace. A chemical etching was used to form an oxide layer with a nanoscale texture on Ti; this layer is hydrophilic, but without hydroxyl groups with high acidic–basic chemical reactivity. The samples were characterized by electron and fluorescence microscopies, UV–Vis spectroscopy, contact angle measurements, zeta potential titration curves, and Folin–Ciocâlteu test. The presence of an adsorbed layer of polyphenols on the functionalized surface, maintaining redox ability, was confirmed by several tests. Consistent with the surface features, the adsorption was maximized by dissolving the extract in a high–amino acid medium, with respect to an inorganic solution, exploiting the high affinity of amino acids for polyphenols and for porous titanium surfaces. The osteogenic differentiation was assessed on an osteoblastic cell line by immunofluorescence, cell viability, expression of key osteoblast markers, and extracellular matrix mineralization. The surfaces functionalized with the extract diluted in the range 1 × 10^−5^–1 mg/mL resulted in having a greater osteogenic activity for the highest concentration, with lower values of cell viability; higher expression of alkaline phosphatase, bone sialoprotein, and collagen; and lower levels of osteopontin. In conclusion, the functionalization of a nano-textured Ti surface with polyphenols can potentially favor the osteogenic activity of osseointegrated implants.

## 1. Introduction

Surface functionalization with biomolecules that are able to favor the osteogenic activity on titanium implants is a challenging aim with a great potential impact on implantology [1]. Three items must be optimized for an effective result: the features of the titanium surface, parameters of the functionalization process, and selection of the biomolecule for the functionalization.

Concerning the surface features of the titanium substrate, the morphology, chemical composition, and reactivity of the surface influence the functionalization process and biological response. The presence of topographical features on the micro- and nanoscale can have several positive outcomes. An incremented roughness is proved to enhance the formation of focal adhesions by osteoblasts through specific integrin recruitment and the response of filopodia and lamellipodia [2], but micro/nanoporosity can also act as a niche for adsorption of proteins and biomolecules [3]. For the present research work, the selected titanium surface was obtained by etching commercially pure titanium in a mixture of sulfuric acid and hydrogen peroxide, with the formation of an oxide layer with a texture on the nanoscale [4]. A chemical surface treatment based on these two reagents has already proved to upregulate the early expression of alkaline phosphatase (ALP), bone sialoprotein (BSP), and osteopontin (OPN) and to enhance bone-like matrix formation in primary osteogenic cell cultures [5]. This research aimed to perform a further chemical and physical characterization of the generated nanotopography finalized to its functionalization for a further improvement of the biological response. In fact, different functionalization processes can be used according to the surface chemical reactivity of the selected substrate. The titanium oxide layer can have different chemical compositions and degrees of hydroxylation according to the chemical environment where it is obtained. The degree and type of hydroxylation of the titanium oxide layer are relevant features for the functionalization: the link between the biomolecules and the surface can occur through an electrostatic attraction if the surface has acidic or basic hydroxyl groups which are deprotonated or protonated at the pH used for the functionalization. These functional groups also affect the bioactivity of the titanium surface, that is, the ability to induce the precipitation of hydroxyapatite in contact with the physiological fluids [6]. Otherwise, physical adsorption can occur if the surface has no functional groups with a strong chemical acidic–basic reactivity.

The parameters to be selected for the functionalization process are the solution’s pH and chemical composition. Bivalent metal ions (such as Ca^2+^) can be added to inorganic aqueous solutions to act as a link between the surface and grafted biomolecule when both have a negative charge and/or the biomolecule makes complex compounds with these metal ions [7]. In some cases, the metal ions are directly released by the substrate to be functionalized if it is reactive in the functionalizing solution; this is the case of hydroxyapatite, for instance [8]. On the other side, solutions rich in organic compounds, such as amino acids and proteins, can be used if the biomolecule to be grafted on the surface of the substrate has a chemical affinity for these compounds, as in the case of polyphenols, and synergic adsorption of different organic compounds can occur. In this research, an inorganic aqueous solution buffered at a neutral pH and added with calcium ions is compared with a high–amino acid medium for the functionalization of Ti-Nano. It is well-known that proteins—and likely their amino acids—can be largely adsorbed on nano-textured titanium surfaces [9], and the idea is to exploit this affinity for an effective functionalization.

As last, the biomolecules for the functionalization must be selected. A natural extract that is rich in several polyphenols was selected for this research. This selection satisfies different criteria. First of all, polyphenols are well-known for different positive biological effects when tested or clinically used as a solution: anti-inflammatory properties, counteraction of the shift toward osteoclastogenesis in bone-loss pathologies, and prevention of the destruction of the gingival connective tissue and alveolar bone in the case of periodontitis of polymicrobial origin [10,11,12,13].

A synergic effect can occur in a natural extract that is a mixture of different phenolic compounds [14]. As it was described in References [7,8], the most abundant compounds in the extract used in this research are the condensed tannins, whereas monomer flavan-3-ols account for only about 5% of the polymeric forms. The monomer units composing the condensed tannins are (+)-catechin, (−)-epicatechin, (−)-epicatechin-3-O-gallate, and (−)-epigallocatechin; (−)- epicatechin is the most abundant monomer unit. The extract has significant antioxidant and radical scavenging action. Polyphenols are not significantly bioavailable when they are assumed through food and go through the intestinal tract; the idea of this research is to include these biomolecules in medical devices through surface functionalization being locally more bioavailable and chemically stable, largely exercising their properties. The extract comes from the by-product of the wine industry (red grape pomace), with responsible use and valorization of the local resources in line with a circular economy approach.

This research contributes to the investigation of whether the process of bone repair might benefit from the paracrine secretion of osteoblasts. The research layout is the chemical and physical characterization of the nano-textured titanium surface, the selection of the best functionalizing solution and parameters, and the evaluation of the osteogenic differentiation on the functionalized surface.

## 2. Materials and Methods

### 2.1. Sample Preparation

Commercially pure Ti discs (grade 2—Realum, São Paulo, SP, Brazil) that were 13 mm in diameter and 2 mm thick were grinded by silicon carbide sandpapers (320, 600) and washed in an ultrasound bath (Ti-MP). Some samples were chemically etched in a solution of equal volumes of concentrated H_2_SO_4_ (95–97%) and 30% H_2_O_2_—100 mL for each of the 10 Ti discs—for 4 h, in a thermostatic ice bath, under stirring, similarly to a published protocol [5]. The discs were then washed in distilled water, autoclaved, and air-dried (Ti-Nano).

The extract of red grape (Barbera) pomace was prepared as described in Reference [8]. For the first process of functionalization, diluted solutions of the extract were prepared in a high–amino acid medium composed of DMEM (Dulbecco’s Modified Eagle’s Medium, Invitrogen, Carlsbad, CA) and 1% penicillin–streptomycin (Sigma-Aldrich) at the concentrations of 1 × 10^−5^ mg/mL, 1 × 10^−3^ mg/mL, and 1 mg/mL of the extract (PPHE). These concentrations were defined based on a pilot cell culture in the presence of diluted extracts that showed enhanced osteogenic differentiation at 1 × 10^−5^ mg/mL.

These solutions were used for the functionalization of the chemically treated Ti samples. The Ti-Nano discs were submerged in 1 mL of each solution and maintained overnight at 37 °C in a humidified atmosphere, containing 5% CO_2_ and 95% atmospheric air. Then the Ti discs were washed with PBS at 37 °C.

Some samples were prepared analogously, but without the addition of the polyphenols extract to the high–amino acid medium (Ti-Nano/HP). Some samples specifically devoted to the analyses in which the presence of phenol red in the high–amino acid medium could give an interference were prepared analogously, but in a DMEM medium without phenol red (Gibco^®^ DMEM without phenol red—A14430-01) and with (Ti-Nano/P_HP1b) or without (Ti-Nano/HPb) the addition of the polyphenols extract.

For the second process of functionalization, the extract was diluted in an inorganic solution composed of the TRIS/HCl buffer (pH 7.4), with the addition of CaCl_2_ (0.292 g/L) with a concentration of 1 mg/mL of extract. The solution was magnetically stirred for 1 h in the dark. For the functionalization, the chemically treated Ti discs were placed in a dark container with 5 mL of the functionalizing solution in an incubator at 37 °C for 3 h. In the end, each sample was washed in ultra-pure water twice, left to dry under a hood, and stored in a multi-well plate in dark conditions (Ti-Nano/P_Ca) [7]. A summary of all the prepared samples is reported in Table 1.

### 2.2. Zeta Potential

The measurement of the zeta potential was carried out on the Ti discs (Ti-MP and Ti-Nano) by using an Eletrockinetic analyzer (Electrokinetic Analyzer/EKA, Anton Paar Gmbh, Graz, Austria). A titration of zeta potential as a function of pH was measured in a 0.001 M KCl solution with an automatic titration unit. For statistical reasons, four measurements were performed at each pH value. Mean and standard deviation were calculated to plot the zeta potential vs. pH curve. Two samples were placed parallelly, with a gap of about 100 μm between them; the electrolyte was forced to flow in the gap; and the zeta potential was measured from the difference in potential measured at the edges of the measuring cell, following the Helmholtz–Smoluchowski equations [15]. The same couple of samples were used both for the acidic and basic range, starting at pH 5.5 in both cases.

### 2.3. Electron and Fluorescence Microscopies

The surface morphology of the samples was observed and analyzed in a GeminiSEM field emission scanning electron microscope (FESEM) (Carl Zeiss, Oberköchen, Germany). The X-ray dispersive energy spectroscopy (EDS) analysis was performed for a semi-quantitative microanalysis of the elements on an area of 200 × 200 micron or on spots on the surface. The polyphenols adsorbed on the functionalized samples were observed through fluorescence microscopy by means of a laser confocal microscope (LSM 900, Zeiss, Dusseldorf, Germany), and the images were processed by the Zen software (Axio Imager 2, Zeiss), exploiting polyphenols autofluorescence [16,17]. A red filter and excitation wavelength (573 nm) were used to observe the red autofluorescence emitted by polyphenols, setting 1 s of exposure time and with a magnification of 200×.

### 2.4. UV–Vis Spectroscopy

The samples were analyzed in reflectance tests by means of a UV–Vis spectrophotometer (UV–Visible Spectrophotometer UV-2600, Shimadzu). An integrating sphere was used to obtain a diffuse reflectance spectrum; this was necessary because the tested samples were not perfectly flat, and they reflected the light in different directions.

### 2.5. Contact Angle Measurements

The wettability of the samples was measured by measuring the static contact angle of a sessile drop. A drop of double-distilled water (5 µL, γ = 72 Mn/m) was deposited on the top of the surface under analysis, and its shape was recorded by the following instrument and software: DSA-100, KRÜSS GmbH, Hamburg, Dropshape Analysis.

### 2.6. Folin–Ciocâlteu Test: Quantification of the Total Phenolic Content

The Folin–Ciocâlteu test quantified the total polyphenols’ content (as Gallic Acid Equivalents, GAE), measuring the absorbance at 760 nm, with a wavelength of blue color resulting from the redox reaction between the Folin–Ciocâlteu reagent (Folin–Ciocâlteu phenol reagent, Sigma-Aldrich), which contains phosphotungstic/phophomolybdic acids, and polyphenols. The measurement was performed through UV–Vis spectroscopy after a 2 h reaction between the sample and the Folin–Ciocâlteu reagent, as previously described [7].

### 2.7. Biological Experiments

#### 2.7.1. Culture of Osteogenic Cells

Cells of the UMR-106 cell line (ATCC^®^ CRL1661™, American Type Culture Collection, Manassas, VA, USA), stored in 2 mL cryogenic tubes submerged in liquid nitrogen, were thawed and then cultured in 75 cm^2^ culture bottles (Corning Inc., New York, NY, USA) with 20 mL Dulbecco’s Modified Eagle Medium (D-MEM, Invitrogen, Carlsbad, CA, USA), 10% fetal bovine serum (Invitrogen), and 1% penicillin–streptomycin (Sigma-Aldrich, St. Louis, MO, USA). During the entire culture period, the cells were kept at 37 °C, in a humidified atmosphere, containing 5% CO_2_ and 95% atmospheric air, and the culture media were changed every 2 days. After 80% confluence, cells were removed from culture flasks by treatment with 1 mM EDTA (Gibco, Grand Island, NY, USA) and 0.25% trypsin (Gibco) and counted in an automatic cell counter (Countess™ Automated Cell Counter, Invitrogen). Cells were then plated on titanium discs in 24-well plates (Corning Inc.), at a density of 10,000 cells/well; cultured in an osteogenic medium consisting of D-MEM (Invitrogen) supplemented with 10% fetal bovine serum (Invitrogen), 2.2 mL penicillin–streptomycin (Sigma-Aldrich), 5 μg/mL ascorbic acid, and 7 mM beta-glycerophosphate; and then kept in a humid atmosphere with 5% CO_2_ and 95% atmospheric air. Cell growth on the polystyrene plates was assessed by observation under the Axiovert 25 inverted-phase microscope (Zeiss Inc., Göttingen, Germany).

#### 2.7.2. Cell Morphology by Epifluorescence Microscopy

On days 3 and 7 of culture, cells were fixed in 4% paraformaldehyde in 0.1 M phosphate buffer (PB) at pH 7.2 for 10 min at room temperature (RT). Then the samples were processed for indirect immunofluorescence as described previously [5]. Permeabilization was performed with 0.5% Triton X-100 solution in PB for 10 min, followed by blocking with 5% skimmed milk in PB for 30 min. For the labeling of UMR-106 cells, primary antibody-to-bone sialoprotein (*Bsp*) (1:200, WVID1-9C5, Developmental Studies Hybridoma Bank—DSHB, Iowa City, IA, USA) was used, followed by Alexa Fluor 594–conjugated secondary antibody (red fluorescence; 1:200, Molecular Probes) in a solution containing Alexa Fluor 488–conjugated phalloidin (green fluorescence; 1:200, Molecular Probes) for the visualization of the actin cytoskeleton. Incubations were performed in a humidified atmosphere for 60 min at RT. Between each incubation step, the samples were washed 3 times—5 min each—in PB. Before mounting for microscopic observation, the cell nuclei were labeled with DAPI (Molecular Probes) at 300 nM for 5 min, and the samples were then washed quickly with deionized water. Ti discs were first mounted on Fisherbrand glass slides, and 12 mm glass coverslips (Fisher Scientific) were mounted on the discs in Vectashield anti-fade mounting medium (Vector Laboratories, Burlingame, CA, USA). The samples were examined by using a Leica DMLB fluorescence microscope (Leica, Bensheim, Germany) that was coupled with a Leica DC 300F digital camera, and the acquired images were processed with the Adobe Photoshop CS5.1 program (Adobe Systems Inc., San Jose, CA, USA).

#### 2.7.3. Cell Metabolic Activity/Cell Viability Assay

On days 3, 7, and 10 of culture, UMR-106 cell metabolic activity/viability was assessed by the MTT {[3-(4,5-dimethylthiazol-2-yl)-2,5-diphenyltetrazolol] bromide} colorimetric assay (Sigma) [18]. Briefly, after removing the culture medium, the cells were incubated with osteogenic medium + 10% MTT for 4 h at 37 °C, in a humidified atmosphere containing 5% CO_2_ and 95% atmospheric air. The culture medium was then removed from the wells, and 1 mL acidic isopropanol solution (Merck, Darmstadt, Germany) was added to each well, under stirring, for 5 min, for complete solubilization of the formed precipitate. Then 150 µL aliquots were transferred to a 96-well plate for reading on a spectrophotometer (µQuanti, BioTek Instruments Inc., Winooski, VT, USA), at a wavelength of 570 nm.

#### 2.7.4. mRNA Expression of Osteogenic Markers by Real-Time Polymerase Chain Reaction (Real-Time PCR)

The mRNA expression of alkaline phosphatase (*Alp*), bone sialoprotein (*Bsp*), collagen (*Col*), osteocalcin (*Oc*), osteopontin (*Opn*), and runt-related transcription factor 2 (*Runx2*) was evaluated by real-time PCR on day 5 of culture. The culture medium was removed from the wells, and 1 mL TRIzol reagent (Invitrogen) was added to the first well to promote cell lysis by homogenization. This mixture was transferred to the next well until the last well of each experimental group. The samples were kept at RT for 15 min and stored at −20 °C for at least 24 h. Then an aliquot of 200 µL chloroform (Merck, Darmstadt, HE, Germany) was added to each TRIzol suspension. The tubes were manually shaken for 30 s and kept on ice for 5 min. The samples were then centrifuged for 15 min, at 4 °C, at 10,500 rpm (AccuSpin Micro R, Fisher Scientific, Pittsburgh, PA, USA), and the aqueous phase (upper) was collected in new 1.5 mL tubes. An aliquot of 250 µL 96% ethanol (Merck) was added to the samples and then centrifuged in silica gel columns present in the SV Total RNA Isolation System kit (Promega, Madison, WI, USA). The sample/ethanol solution was gently stirred and transferred to a new silica gel column. Specific buffers were added and interspersed with brief centrifugations of 15 s at 10,500 rpm each. After several washes with different buffers, the ribonucleic acid (RNA) samples were eluted in RNAse-free water and stored at −80 °C until complementary DNA (cDNA) was made. Then total RNA was quantified at different wavelengths (260, 280, 230, and 320 nm) in a GeneQuant 1300 spectrophotometer (GE Healthcare, Fairfield, CT, USA). Its integrity was assessed by using the Agilent 2100 BioAnalyzer instrument (Agilent Technologies, Santa Clara, CA, USA). RNA integrity was verified from 100 ng total RNA, following the manufacturer’s instructions, and RIN (RNA integrity number) values greater than 8 were considered adequate [19,20]. An aliquot of total RNA equivalent to 1 µg was used for cDNA construction by means of the High-Capacity cDNA Reverse Transcription kit (Applied Biosytems, Foster City, CA, USA), according to the manufacturer’s instructions. In a 200 µL tube, 1 µg total RNA was diluted in a final volume of 10 µL RNAse-free, 2 µL reverse transcriptase buffer, 0.8 µL deoxynucleotide (dNTP), 2 µL random primer, 1 µL MultiScribe™ reverse transcriptase, 1 µL RNAse inhibitor, and 3.2 µL DEPC water (Acros Organics, Waltham, MA, USA), for a final volume of 20 µL/reaction. Then the samples were incubated in a MasterCycler Gradient Thermal Cycler (Eppendorf AG, Hamburg, Germany) at 25 °C for 10 min, 37 °C for 120 min, 85 °C for 5 min, and then cooled to 4 °C. At the end of the reverse-transcription reaction, the cDNA samples were stored at −20 °C. For the real-time PCR reaction, TaqMan^®^ probes (Applied Biosystems) were used for the genes *Alp*, *Bsp*, *Col*, *Oc*, *Opn*, and *Runx2* (Table 2), and the reactions were performed in the StepOne Plus device (Life Technologies, Carlsbad, CA, USA). Reactions were performed in triplicate, using 5 μL Taqman^®^ Gene Expression Master Mix, 0.5 μL Taqman^®^ Gene Expression Master Mix (Probes + Primers) and 4.5 μL cDNA (11.25 ng), for a final volume of 10 μL/reaction. Amplification reactions consisted of 2 min at 50 °C, 10 min at 95 °C, forty cycles of 15 s at 95 °C, and 1 min at 60 °C (denaturation and extension). The results were analyzed based on the Ct value (threshold cycle), which corresponds to the number of cycles in which the amplification of the samples reaches a threshold (determined between the fluorescence level of the negative controls and the phase of exponential amplification of the samples) that allows the quantitative analysis of the expression of the evaluated factor. Expression of the glyceraldehyde-3-phosphate dehydrogenase (GAPDH) gene was used as an endogenous control. A negative sample (water) was subjected to reaction with each Taqman^®^ probe used. The comparative method of 2^−ΔΔCt^ was used to compare the gene expression of cells from different experimental groups [21].

#### 2.7.5. Mineralized Matrix Formation

On days 7 and 10 of culture, UMR-106 cells were washed in Hanks’ solution, fixed in 70% ethanol at 4 °C for 1 h, and then washed in PBS and deionized water. The cultures were then stained with 2% Alizarin Red, pH 4.2, at RT for 15 min, washed in PBS and deionized, and allowed to dry at RT. Images were obtained digitally with a Canon EOS Digital Rebel camera, 6.3 Megapixel CMOS sensor, with Canon EF 100 mm f/2.8 macro lens (Canon, Lake Success, NY, USA). Quantitative analysis of calcium accumulation was performed by using the method described in Reference [22]. In each well, 280 µL of 10% acetic acid was added, and the culture plates were shaken for 30 min. The cell layer was then scraped off, and the solution was transferred to 1.5 mL tubes to be readily vortexed for 30 s. The samples were heated at 85 °C for 10 min, cooled on ice for 5 min, and centrifuged at 13,000 rpm for 20 min. Then 100 µL of supernatant from each sample was transferred to each well of the 96-well plate, and 40 µL of 10% ammonium hydroxide was added. The plates were read in a spectrophotometer (µQuanti), at a wavelength of 405 nm.

#### 2.7.6. Statistical Analysis

The quantitative data obtained from the cell culture experiments were evaluated by one-way or two-way analysis of variance (ANOVA), followed by the Tukey’s post hoc test when appropriate. The significance level was set at 5%.

A list of the analyzed samples is reported in Table 1.

## 3. Results

### 3.1. Chemical–Physical Characterization

The effect of two different media for dissolving polyphenols in the functionalization process of a nano-textured titanium surface was investigated and discussed by considering the chemical and physical features of the surface: an inorganic buffered solution (containing TRIS/HCl and CaCl_2_) and high–amino acid medium were compared. The inorganic solution was selected because it is buffered at pH = 7.4; at this pH, polyphenols have deprotonated acidic OH groups, and they have a negative charge. In the presence of Ca^2+^ ions in the functionalizing solution, the deprotonated polyphenols form complexes of organic compounds with calcium ions, and these complexes can facilitate the grafting on the titanium surface [7]. This process has proved to be effective when the surface of Ti shows a negative charge to link the calcium ions [7]. A high–amino acid medium was also tested here: in this case, in addition to neutral pH and the presence of calcium ions, the chemical affinity of amino acids for polyphenols on one side and for a porous titanium surface on the other one can be exploited for functionalization [3,23]. In this case, the presence of negatively charged functional groups on the surface is not needed for linking polyphenols through physical adsorption.

A detailed investigation of the surface features is needed to predict the adsorption mechanism and to correctly select the process for the surface functionalization. The chemical–physical characterization of the samples was validated by FESEM–EDS, fluorescence microscopy, UV–Vis spectroscopy, contact angle measurements, zeta potential titration curves, and the Folin–Ciocâlteu test. The samples functionalized with the highest concentration of polyphenols were selected for the chemical–physical characterization. Data by Fourier-Transform InfraRed spectroscopy (FTIR) are reported in the Appendix A because they give some information; however, they are not strictly relevant for the main purpose of the paper.

#### 3.1.1. Zeta Potential Titrations

The zeta-potential titration curves allowed us to measure the isoelectric point and investigate the eventual change of the exposed surface functional groups after the chemical treatment of the Ti surface (Figure 1).

The curves obtained on Ti-MP and Ti-Nano are very similar, with only small differences, which may be attributed to the formation of a surface oxide layer during the chemical treatment. The isoelectric point of the two curves is the same, and it is close to 4. An isoelectric point close to 4 is expected for a surface without functional groups able to be protonated or deprotonated and consequently able to influence the surface charge and zeta potential [15]. It means that, both before and after the chemical etching, no hydroxyl groups are acting like a strong acid or base on these surfaces. The titration curve of Ti-Nano has a slightly lower slope than Ti-MP; it can be explained by a greater hydroxylation and hydrophilicity of the oxide layer, inducing lower adsorption of ions (H_3_O^+^ or OH^−^) from the solution during the titration, because the water molecules are more strongly adsorbed [15]. Altogether, the titration curves suggest that hydroxyl groups are present on the surface of Ti-Nano, but they do not have a strong acidic–basic reactivity. The presence of a small plateau at a pH higher than 8 in the curve of Ti-Nano can be explained by the presence of a few acidic OH groups: they act as a very weak acid and are completely deprotonated only at a high pH. This means that Ti-Nano does not expose any protonated/deprotonated functional groups at the pH of functionalization (neutral), and the adsorption of polyphenols or amino acids cannot occur through a strong electrostatic interaction. It can be also noted that the standard deviation registered on the Ti-Nano sample is generally low; this means that the oxide layer is chemically stable in the range of pH considered in the titration. This is of relevance for good protection against corrosion phenomena.

#### 3.1.2. FESEM and EDS Analysis

The surface morphology of the observed samples is shown in Figure 2. The polished Ti surface (Ti-MP) reveals only scratch lines from the polishing treatment. There is no surface feature at the nanoscale on these samples (Figure 2A,B).

The surface of the chemically treated sample (Ti-Nano) is significantly different from that of the polished one when observed at high magnification. A nanoporous morphology appears (Figure 2E) with pores that are 10–25 nm in diameter, similar to a previous observation [4]. This difference is not evident at low (equal or minor to 10,000×) magnification (Figure 2D).

After the functionalization process with polyphenols dispersed in an inorganic solution (Ti-Nano/P_Ca) or a high–amino acid medium (Ti-Nano/P_HP1), agglomerates (between 1 and 8 µm) are observable in both cases on the surfaces. The number of agglomerates is quite larger when the functionalization is made in a high–amino acid medium (Figure 2L,M); in this case, the surface is almost completely covered by the agglomerates when observed at low magnification. On the other side, separate and randomly dispersed agglomerates are observable on Ti-Nano/P_Ca (Figure 2G,H).

EDS analysis was performed in two different modes: on an area (200 × 200 micron) to analyze the average chemical (elemental) composition of the surfaces on all the samples; and on some spots on Ti-Nano/P_Ca and Ti-Nano/P_HP1, focused on the aggregates in order to investigate their specific chemical composition.

As shown in Figure 2C–F, the reference metal surfaces (Ti-MP and Ti-Nano) are only composed of three elements (carbon, oxygen, and titanium). Concerning the difference between the surface chemical composition of Ti-MP and Ti-Nano, almost no change is observed, except for a slight increase of the oxygen content in the latter one, as expected, considering the formation of a surface oxide layer thicker than the native one during the chemical treatment. The detected amount of oxygen is low because the EDS analysis has a penetration depth on the micrometric scale, and it is expected to be quite larger than the thickness of the native or chemically induced surface oxide layer. The amount of carbon detected is as low as unavoidable organic contamination allows on a titanium surface.

In contrast, when the same comparison is made between Ti-Nano (Figure 2F) and Ti-Nano/P_HP1 or Ti-Nano/P_Ca, a consistent increase of C can be observed (Figure 2I,L–N). The aggregates clearly visible on the functionalized surfaces were revealed to be almost completely composed of C (98%) when a spot EDS analysis was performed on them.

#### 3.1.3. Fluorescence

The distribution of polyphenols adsorbed on the different explored surfaces was optically visualized by fluorescence microscopy, exploiting the auto-fluorescence of these biomolecules, as previously reported in the literature [24]. The high–amino acid medium used for the preparation of Ti-Nano/P_HP1 is enriched with a significant amount of phenol red; this means that a surface functionalized in this medium can show a strong fluorescence signal due to adsorption of this fluorescent compound. Hence, a medium without phenol red was hence used for the preparation of the samples for fluorescence microscopy to avoid any interference and to observe the presence of the adsorbed polyphenols on the surfaces. The fluorescence images of the Ti nanostructured surface (Ti-Nano) and the Ti nanostructured sample soaked in the high–amino acid medium without phenol red (Ti-Nano/HPb) were here used as control surfaces. No significant fluorescent signal is observable on these surfaces, as expected (Figure 3). On the other side, Ti-Nano/P_HP1b shows a marked fluorescent signal, thus confirming the presence of polyphenols molecules on the titanium surface. They appear like micrometric agglomerates. This is expected because polyphenols can aggregate by interacting with the amino acids of the medium, and this agrees with the FESEM images.

The sample functionalized in an inorganic solution (Ti-Nano/P_Ca) shows, in the fluorescence images, a significantly lower amount of polyphenols, but some isolated aggregates are observable, and this is in agreement with the FESEM images.

#### 3.1.4. UV–Vis

The presence of the adsorbed layer of biomolecules was analyzed also by UV–Vis spectroscopy (Figure 4). The following samples were compared: Ti-MP, Ti-Nano, Ti-Nano/HP, Ti-Nano/P_Ca, and Ti-Nano/P_HP1. Analogous results were obtained on Ti-Nano/HPb and Ti-Nano/P_HP1b but are not reported here.

The polished titanium metal (Ti-MP) has, as expected, a high reflectance both in the near-UV (200–380 nm) and in the visible (380–780 nm) wavelength ranges, with a broad absorption in the near UV. Looking at Figure 4, we can see that the detected spectrum is close to that reported in the literature for TiO_2_ [25] because the polished Ti surface is covered by a thin layer of titania (native oxide). As expected, the Ti-Nano sample has a lower reflectance than the polished surface in all the ranges of wavelengths; this effect is due to the formation of a thin oxide layer on the surface of the sample during the chemical treatment.

When the metal surface is covered by organic compounds, such as amino acids and/or polyphenols, reflectance is expected to be further reduced on all the explored range of wavelengths; moreover, a specific absorption (less reflectance) can be expected at the wavelength corresponding to the specific electronic transitions of the functional groups typical of the adsorbed organic compounds.

Ti_Nano/HP has a spectrum with even lower reflectance than Ti_Nano because a layer of amino acids from the medium is adsorbed on the surface oxide layer due to the soaking in the high–amino acid medium.

The results show that the spectrum of Ti-Nano/P_HP1 does not superpose the Ti-Nano/HP one. Ti-Nano/P_HP1 has a further lower reflectance and a different shape of the curve; this effect confirms the presence of polyphenols on the surface, which act as a shield against the UV–Vis light. Moreover, it must be noted that phenolic acids and flavonoids have a specific absorption in the range of 270–330 nm; hydroxycinnamic acid adsorbs at 277–280 and 313–330 nm, with a shoulder at ~290 nm; and anthocyanidins adsorb at 280–320 nm [26]. The formation of a plateau with low intensity in the range from 200 to 330 nm in the spectrum of Ti-Nano/P_HP1 can be ascribed to the specific absorption of polyphenols in this range. The slope of the curve around 350 nm is related to the charge transfer from the valence to the conduction electronic bands of the titanium oxide layer, and it changes after polyphenols adsorption.

A spectrum is also acquired on the sample functionalized in an aqueous solution (Ti-Nano/P_Ca). This spectrum is different from that of the nano-textured titanium (Ti-Nano) in the near UV region, while it becomes like that of Ti-MP in the range of visible light. This effect can be explained by considering that, in absence of amino acids, proteins, and organic compounds in the functionalizing solution, polyphenols are deposited on the surface of Ti-Nano as separated agglomerates, but there is no layer of adsorbed compounds almost completely covering the surface, and that is why high reflectance of the visible light is maintained. On the other side, absorption occurs in the UV range because of the presence of polyphenols. This result agrees with the observations made by fluorescence microscopy.

#### 3.1.5. Contact Angle

Despite the biological relevance of surface wettability, the contact angle measurements can be used as a method to confirm the presence of organic compounds adsorbed on a metal surface. Ti-MP and Ti-Nano were used as a control (Table 3).

The chemical treatment of the Ti surface significantly decreases the water contact angle (from 70° on Ti-MP to 31° on Ti-Nano) and increases hydrophilicity, as compared to Ti-MP. The previously reported zeta-potential titration curve of Ti-Nano indicates a higher wettability of this surface with respect to Ti-MP, as is in agreement with this result.

The presence of amino acids precipitated on the surface from the high–amino acid medium induces a further decrease of the contact angle down to 10° (Ti-Nano/HP).

The presence of polyphenols, on the other side, has a small hydrophobic effect (the contact angle is around 40° on Ti-Nano/P_HP1). The difference with respect to Ti-Nano/HP confirms that polyphenols are also deposited during the functionalization processes in the high–amino acid culture medium. Polyphenols are amphiphilic, with a hydrophobic organic chain and hydrophilic functional groups (OH); the results evidence the exposition of the organic backbone of the biomolecule after adsorption.

The same effect of polyphenols of increasing hydrophobicity is observed when the functionalization is performed by using an inorganic solution and the contact angle is significantly increased on Ti-Nano/P_Ca, with respect to Ti-Nano, up to 37°. The similarity between the contact angles obtained on Ti-Nano/P_Ca and Ti-Nano/P_HP1 supports the hypothesis that this specific angle is related to the wettability of the grafted polyphenols.

#### 3.1.6. Redox Activity and Total Phenolic Content (Folin–Ciocâlteu Test)

The Folin–Ciocâlteu method is applied to evaluate if the polyphenols adsorbed on the different samples are present and have the ability to act as reducing agents. The method, originally developed to analyze polyphenols in a solution, was here modified to be applied to solid samples and polyphenols adsorbed on a surface. The unfunctionalized substrate (Ti-Nano), Ti-Nano/HP, and Ti-Nano/HPb were here used as control surfaces. Ti-Nano/HP involves the redox ability of the phenol red compound coming from the high–amino acid medium, and this positive signal must be not confused with that of the polyphenols. Ti-Nano/HPb allows us to verify if any redox signal comes from the adsorbed amino acids. Table 4 shows the detected signals, expressed as GAE units; it means that an analogous redox activity is detected in a solution of that concentration of gallic acid, which is a simple polyphenol commonly often used as a model molecule.

As expected, the redox activity was null on Ti-Nano and Ti-Nano/HPb, and this was either because no compound is adsorbed (Ti-Nano) or none of the compounds that were adsorbed has redox ability (Ti-Nano/HPb). The null signal on Ti-Nano/HPb confirms that the amino acids adsorbed from the used medium do not any redox ability. A positive signal was detected on Ti-Nano/HP because of the adsorption of phenol red.

The positive signal detected on Ti-Nano/P_Ca can be ascribed to the adsorbed polyphenols: they maintain their redox ability also after adsorption on the titanium surface. The detected signal on Ti-Nano/P_HP1 is due both to the adsorbed polyphenols and phenol red, and, as expected, it is the highest detected signal. If the GAE value detected on Ti-Nano/HP (due to phenol red) is subtracted from that of Ti-Nano/P_HP, the obtained redox ability (due only to the polyphenols) is, in any case, significantly higher than that of Ti-Nano/P_Ca, and this is in agreement with the larger adsorption of polyphenols detected by the other techniques on Ti-Nano/P_HP1.

### 3.2. Biological Experiments

According to the chemical–physical characterization, the biological experiments were performed only on the samples functionalized in the high–amino acid medium, as they are more promising, and Ti-Nano was used as a control surface for all biological tests. Different amounts of polyphenols were added to the functionalizing solutions (Ti-Nano/P_HP-5, Ti-Nano/P_HP-3, Ti-Nano/P_HP1) to verify the biological effects of polyphenols, besides the presence of other compounds—supposedly mostly amino acids—adsorbed during the functionalization process.

#### 3.2.1. Cell Morphology by Fluorescence Microscopy

The test was performed by using the UMR-106 cell line that is composed of cells that are already committed to osteoblastic differentiation, according to the approach described in References [27,28].

The epifluorescence analysis revealed that UMR-106 cells adhered and spread on the various functionalized Ti-Nano surfaces and on their control surface (Figure 5), exhibiting on day 3 of culture polygonal morphologies and cell–cell interactions throughout the monolayer (Figure 5A–D). Most cells were labeled for perinuclear BSP and, occasionally, the labeling was also localized extracellularly (Figure 5B). As the culture progressed to day 7, the cells reached confluence (Figure 5E–H), and BSP-positive areas revealed the formation of mineralization nodules (Figure 5I–L). No important differences were observed between the groups.

#### 3.2.2. Cell Metabolic Activity/Cell Viability Assay

The MTT assay was performed on days 3, 7, and 10 of culture (Figure 6). The highest values of cell viability were detected on day 7 when compared with day 3 (at least two times higher), and these remained unaltered during the mineralization phase until day 10 of culture. While there are no differences between groups on day 3, the lowest MTT values were detected in cultures grown on the functionalized surface with 1 mg/mL on days 7 and 10, although they are statistically similar to some of the other groups.

#### 3.2.3. mRNA Expression of Osteogenic Markers by Real-Time PCR

The *Alp*, *Bsp*, and *Col* mRNA expression levels increased in UMR-106 cells grown on Ti-Nano functionalized with 1 mg/mL extract (significantly for *Alp* and *Col*), whereas those of *Oc* and *Opn* reduced (significantly for the latter). *Runx2* expression levels are significantly higher in cultures grown on the functionalized surfaces, except for the 1 × 10^−3^ mg/mL PPHE (Figure 7).

#### 3.2.4. Mineralized Matrix Formation

Alizarin red staining reveals that UMR-106 cell cultures grown on Ti-Nano from all groups exhibited an osteogenic phenotype on days 7 and 10 of culture, with no significant differences between groups in each time point (Figure 8). Despite that, the cultures grown on Ti-Nano functionalized with 1 mg/mL extract show a tendency toward greater calcium content on day 7 of culture (Figure 8 and Appendix A).

## 4. Discussion

Although different strategies for modifying the surfaces of metallic implants have been developed and applied over the last decades, with very satisfactory pre-clinical and clinical results, especially in anatomical sites of low bone quality [29,30], more recent studies have further explored fine-tuning—the surface structuring at the nanoscale—altering its topographical, chemical, and surface energy characteristics. These physical-chemical modifications have been shown to exert, per se, major effects on cells during tissue repair, and might favor the functionalization of the bioactive molecule(s)—for example, with an increase in the surface area—aiming at the control and/or modulation of key biological interfacial events [31,32,33,34,35]). In this context, here we opted to use a nanotopography created by the treatment of Ti surface with a mixture of H_2_SO_4_ and H_2_O_2_, with the aim of functionalizing on its surface a grape-pomace extract with a high concentration of PPHE rich in epicatechin [9]. The stimulatory effects on bone cells and bone repair of a surface treated with a mixture of an H_2_SO_4_ and H_2_O_2_ have been described in in vitro and in vivo exploratory and mechanistic studies by other research groups [36,37,38,39].

Ti-Nano has a nano-textured surface with higher wettability than Ti-MP (contact angle measurements and zeta potential titration curve). This increase in water wettability on the nanostructured surface can be related both to an increase in the surface area by nano-texture, as previously observed in the literature [40], and/or to an increment in the surface density of hydroxyl groups, according to the zeta potential measurement that is not sensitive to topographical features. These groups do not have a strong acidic–basic reactivity, so they contribute to wettability, but not to the presence of a net electrostatic charge on the surface in contact with a liquid (zeta potential titration measurement).

Although in a pilot study that used polystyrene as the substrate, more significant osteogenic effects were observed when cultures were grown in a culture medium supplemented with the grape-pomace extract at a concentration ranging from 0.001 to 0.1 μg/mL, we chose solutions at 0.01 μg/mL, 1 μg/mL, and 1 mg/mL for the functionalization of Ti-Nano. For the chemical–physical characterization of the functionalized surface, the samples obtained with the most concentrated solution (1 mg/mL) were selected, thus ensuring the detectability of the organic coating.

Even if the exact amount of organic components effectively adsorbed from the 1 mg/mL solution of the extract has not been quantified, the surface characterization results showed that it varies depending on the solvent used, whether organic or inorganic. Precipitation and adsorption of organic biomolecules occur on the nano-textured titanium surface by using both the inorganic solution (Ti-Nano/P_Ca) and high–amino acid medium, but it is quite larger in the case of the latter one (Ti-Nano/P_HP1). The adsorbed biomolecules are in the form of aggregates; they are randomly isolated when an inorganic solution is used, while they form an almost continuous layer of aggregates when the functionalization occurs in the high–amino acid medium. Data from FESEM, EDS, fluorescence microscopy, and UV–Vis spectroscopy agree with these conclusions. The absence of deprotonated or protonated functional groups (e.g., the hydroxyl ones) on Ti-Nano allows us to assume a mechanism of physical adsorption and explains the lower efficacy of the functionalization process in the inorganic solution, wherein an electrostatic attraction between the Ca^2+^ ions and the organic compounds is expected. Different mechanisms of functionalization occur when the titanium surface has different chemical features, as reported in the literature. An effective functionalization in an inorganic solution can be obtained on a nano-textured titanium oxide layer with a high density of acidic OH groups with the formation of a continuous layer of grafted polyphenols through the link of calcium ions [7].

The precipitation of other organic compounds, together with the polyphenols, occurs when the functionalization is performed in the high–amino acid medium. These compounds can reasonably be supposed to be mostly amino acids, considering the high affinity of the polyphenols for amino acids and their high concentration in the medium [41], even if the presence of other organic compounds cannot be excluded, at this stage.

Considering the presence of 10% fetal bovine serum in the cell experiments and the high protein content of the physiological fluids, the deposition of organic compounds on the PPHE-functionalized Ti-Nano could be further potentiated, as amino acids and proteins establish multiple non-covalent stabilizing interactions with PPHE, resulting in three-dimensional formations on the scale of nanoparticles [42]. Associated with the reduction of the high hydrophilicity of Ti-Nano detected with the presence of organic coating, these phenomena should affect the described mechanisms of osteoblastic cell interactions with the surface nanotopographic features [43,44].

The results of surface characterization showed that PPHE maintains their antioxidant capacity on the Ti-Nano surface, with which they probably established physisorption. These are crucial aspects that control cellular activity in the interfacial region. While the presence of PPHE with antioxidant activity should limit the apoptosis of osteoblastic cells induced by oxidative stress [45] and favor osteoblastic differentiation via the *Runx2* transcription factor [13], its weak interaction with the nanotopographic TiO_2_ layer could result in a major initial burst release depending on the interstitial fluid components present in the interfacial region, with a potential impact on the sustained desorption process. If these phenomena took place under the experimental conditions used, the availability of PPHE in the extracellular milieu in a static cell culture environment promoted some beneficial effects on osteoblast differentiation and activity.

Concerning the biological tests, the UMR-106 cell line was selected because these cells are in an advanced stage of osteoblastic differentiation and, thus, they are capable of producing a collagen matrix that undergoes mineralization in a shorter culture period: it means, on days 5–7 of culture, that is, 1 to 2 days after the confluence of the culture in our conditions.

The results of the cell-culture experiments showed that, despite the unequivocal osteogenic potential of UMR-106 cells on all studied surfaces, as was detected by the qualitative and quantitative analyses of Alizarin red staining (for the detection of calcium deposits), it was the PCR method that allowed us to detect changes in the osteoblast markers’ expression during the cells grown on Ti-Nano functionalized with 1 mg/mL extract at day 5 of culture, when full osteoblast differentiation is achieved.

Although significant osteogenic differentiation is observed in all groups, the lowest values of cell viability; higher *Alp*, *Bsp*, and *Col* (significantly for *Alp* and *Col*); and lower *Oc* and *Opn* (significantly for *Opn*) mRNA expression levels are detected in the cultures grown on Ti-Nano functionalized with the 1 mg/mL grape-pomace extract solution, suggesting a greater stimulus and/or acceleration of osteoblastic activity on this surface, and it could be related to the relatively higher concentration of PPHE in the extracellular milieu available to cells.

Indeed, PPHE exerts stimulatory effects on osteoblast differentiation and function by promoting *Runx2* expression through different signaling pathways and inducing the expression of genes related to bone-matrix formation, such as *Bsp* (discussed in Reference [11]). We opted to detect bone sialoprotein (BSP) because it is a potent nucleator of apatite crystals that is associated with alkaline phosphatase in biomineralization foci [46]. We selected two time-points: day 3 of culture is during its proliferative phase, and day 7 is during its mineralization phase. Because of its restricted expression, BSP provides a valuable marker for osteoblast differentiation and bone formation [47]. Indeed, the expression of *Alp*, *Bsp*, and *Col* mRNA is crucial for bone-matrix production and mineralization. These are downstream genes that are regulated by *Runx2* [48], which, in turn, has its expression modulated by polyphenols [13]. The observed downregulation of *Opn*, which is expected to occur when osteoblasts achieve full differentiation, also supports this interpretation. As last, an upregulation of OPN and OC—considered late osteoblast markers and inhibitors of matrix mineralization—would be expected to occur at later time points (not determined), during the mineralization phase of cultures [48,49,50].

Despite that, the phenotypic effects on the formation of a mineralized matrix by the presence of PPHE are limited, with only a tendency toward higher calcium content. As other cell types, such as macrophages, might have their functions modulated by PPHE [51] and have crucial roles in the repair process in the interfacial region of biomaterials with the bone tissue, with major effects on the peri-implant tissue phenotype [52], the effective relevance of functionalizing nanostructured Ti surfaces with grape-pomace extract should be further evaluated; strategies and tools for deciphering the reciprocal interactions between osteoblasts and macrophages in in vitro and in vivo models must be used for this purpose.

Further in vitro and vivo studies should verify whether the process of bone repair might benefit from the paracrine secretion/signaling of some cell types other than bone cells, such as macrophages, mast cells, and fibroblasts, whose functions are also modulated by polyphenols.

## 5. Conclusions

The functionalization of Ti-Nano with a solution of grape-pomace extract at 1 mg/mL allows for the availability of PPHE in varying amounts and for the distribution, depending on the solvent used, to be more abundant and less inhomogeneous with the culture medium used for the growth of osteoblastic cells; PPHE functionalized on Ti-Nano maintains its antioxidant activity and interacts with the TiO_2_ nanotopography through physisorption. The in vitro interaction of osteoblastic cells with Ti-Nano functionalized with a solution of grape-pomace extract at 0.01 μg/mL, 1 μg/mL, or 1 mg/mL allows the formation of a mineralized matrix in all groups, with a tendency toward higher osteogenic activity for the 1 mg/mL concentration, with a higher *Alp* and *Col* and lower *Opn* mRNA expression levels. In conclusion, an unequivocal osteogenic potential of UMR-106 cells was qualitatively and quantitatively detected on all studied surfaces, but more advanced osteoblast differentiation was achieved on Ti-Nano functionalized with 1 mg/mL extract at day 5 of culture.

## Figures and Tables

**Figure 1 nanomaterials-12-02916-f001:**
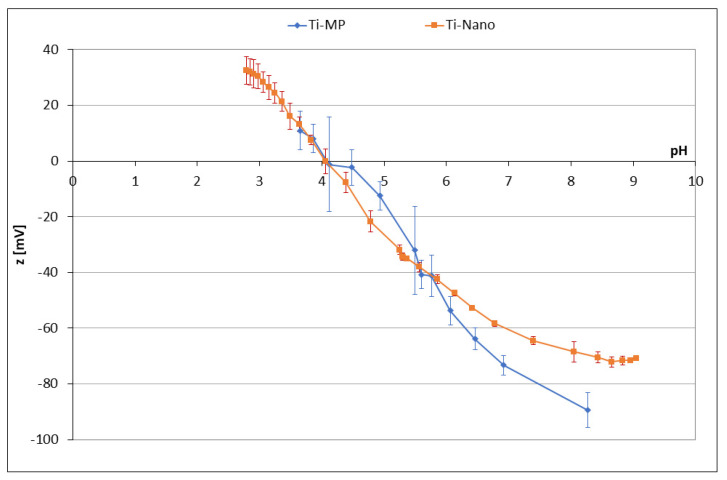
Zeta potential versus pH titrations of Ti-MP and Ti-Nano.

**Figure 2 nanomaterials-12-02916-f002:**
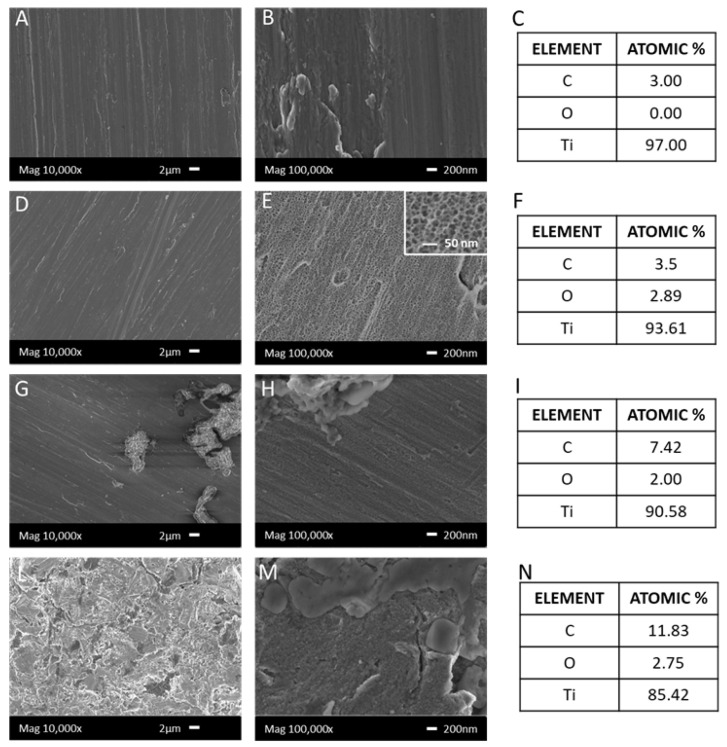
FESEM micrographs and EDS analyses of Ti-MP (**A**–**C**), Ti-Nano (**D**–**F**), Ti-Nano/P_Ca (**G**–**I**), and Ti-Nano/P_HP1 (**L**–**N**), showing the morphology and composition of the surfaces.

**Figure 3 nanomaterials-12-02916-f003:**
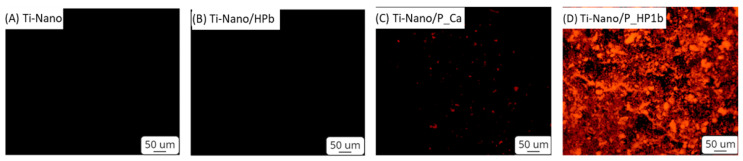
Fluorescence images of Ti-Nano (**A**), Ti-Nano/HPb (**B**), Ti-Nano/P_Ca (**C**), and Ti-Nano/P_HP1b (**D**); filter: 540 nm.

**Figure 4 nanomaterials-12-02916-f004:**
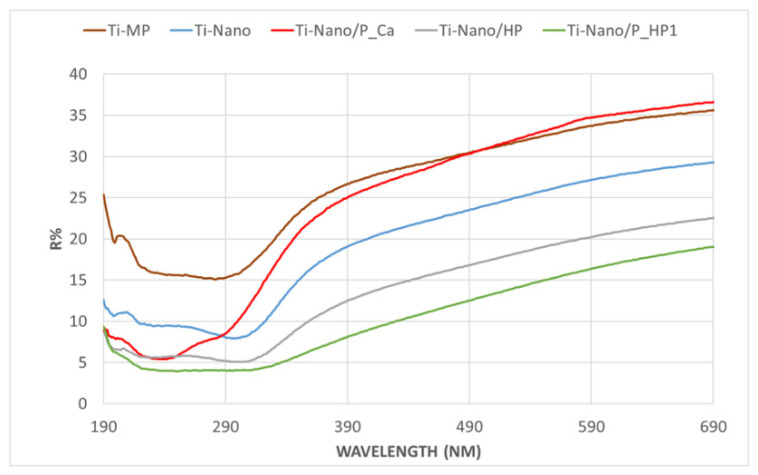
UV–Vis spectra of Ti-MP, Ti-Nano, Ti-Nano/HP, Ti-Nano/P_Ca, and Ti-Nano/P_HP1.

**Figure 5 nanomaterials-12-02916-f005:**
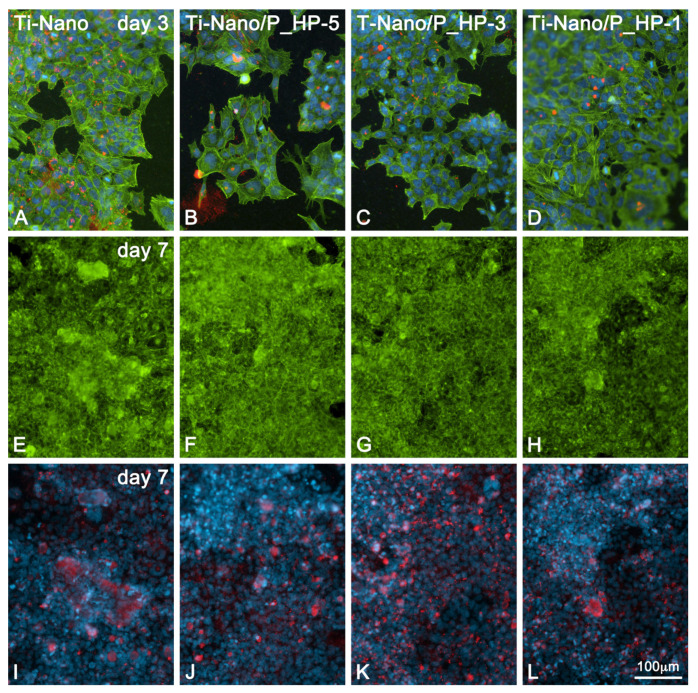
Epifluorescence of UMR-106 cells grown on Ti-Nano surfaces functionalized with 1 × 10^−5^, 1 × 10^−3^, and 1 mg/mL grape-pomace extract and the control surface on days 3 (**A**–**D**) and 7 (**E**–**L**) of culture. (Ti-Nano, Ti-Nano/P_HP-5, Ti-Nano/P_HP-3, and Ti-Nano/P_HP1) Green, red, and blue fluorescence depicts actin cytoskeleton, bone sialoprotein (BSP), and cell nuclei, respectively. Note the BSP-positive mineralized matrix foci in (**I**–**L**). The microscopic fields are the same respectively for (**E**-**I**, **F**-**J**, **G**-**K**, **H**-**L**).

**Figure 6 nanomaterials-12-02916-f006:**
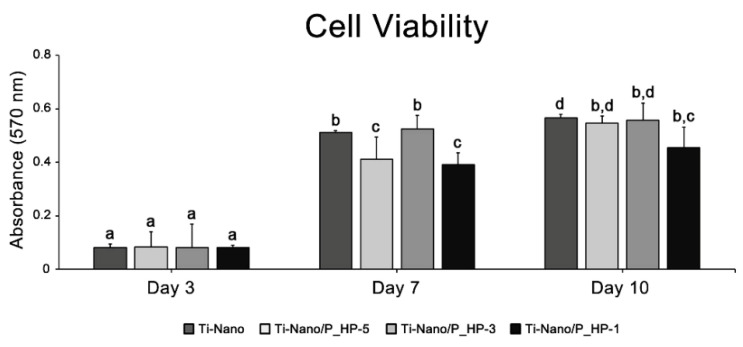
Cell viability (MTT assay, optical density) on days 3, 7, and 10 of UMR-106 cell cultures grown on Ti-Nano surfaces functionalized with 1 × 10^−5^, 1 × 10^−3^, 1 mg/mL grape-pomace extract and the control surface (Ti-Nano, Ti-Nano/P_HP-5, Ti-Nano/P_HP-3, and Ti-Nano/P_HP1). The bars that share the same letter are not significantly different from each other (*p* > 0.05).

**Figure 7 nanomaterials-12-02916-f007:**
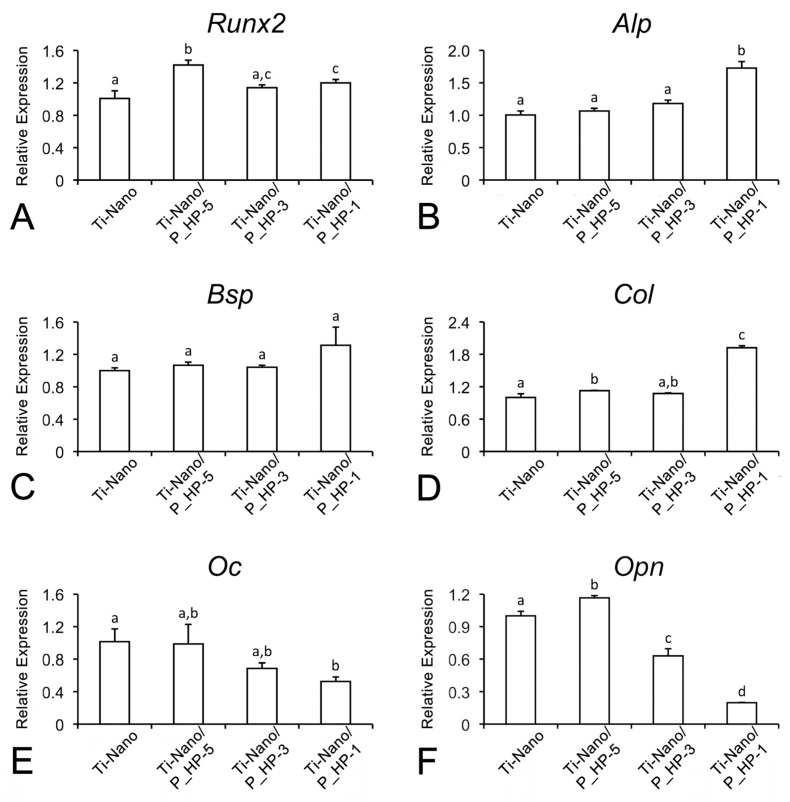
Relative mRNA expression for *Runx2* (**A**), *Alp* (**B**), *Bsp* (**C**), *Col* (**D**), *Oc* (**E**), and *Opn* (**F**) in UMR-106 cells grown on Ti-Nano surfaces functionalized with 1×10^−5^, 1×10^−3^, 1 mg/mL grape-pomace extract and the control surface (Ti-Nano, Ti-Nano/P_HP-5, Ti-Nano/P_HP-3, and Ti-Nano/P_HP1) on day 5 of culture. Data were normalized to *Gapdh*, and the value 1 was assigned to the control group. The bars that share the same letter are not significantly different from each other (*p* > 0.05).

**Figure 8 nanomaterials-12-02916-f008:**
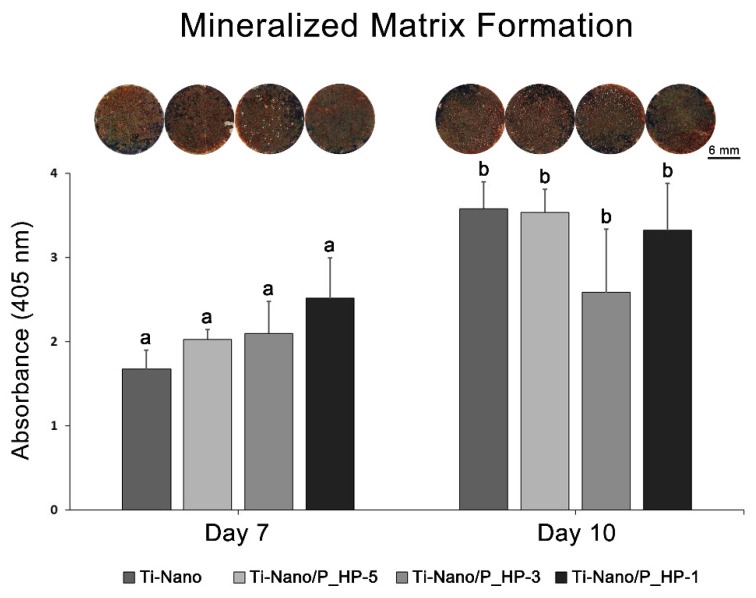
Macroscopic imaging and Alizarin red extraction of UMR-106 cell cultures grown on Ti-Nano functionalized with 1 × 10^−5^, 1 × 10^−3^, 1 mg/mL grape-pomace extract and the control surface (Ti-Nano, Ti-Nano/P_HP-5, Ti-Nano/P_HP-3, Ti-Nano/P_HP1) on days 7 and 10 of culture. Cultures were stained with Alizarin red for the detection of calcium deposits in areas of mineralized matrix formation, which were randomly distributed throughout the entire disk surface (scale bar = 6 mm). The 1 mg/mL group showed a tendency toward greater calcium content on day 7 of culture. Bars that share the same letter are not significantly different from each other (*p* > 0.05). A comparison between periods was not performed.

**Table 1 nanomaterials-12-02916-t001:** Codification and description of the samples.

Surface Treatment					Sample Acronym
Polishing	Chemical Etching (Nano-Texture)	Adsorption of Polyphenols in an Inorganic Solution	Adsorption of Polyphenols in a High–Amino Acid Medium	Soaking in the High–Amino Acid Medium	
X					Ti-MP
X	X				Ti-Nano
X	X			X	Ti-Nano/HP Ti-Nano/HPb (no phenol red)
X	X	X (1 mg/mL)			Ti-Nano/P_Ca
X	X		X (1 mg/mL)		Ti-Nano/P_HP1 Ti-Nano/P_HP1b (no phenol red)
X	X		X (1 × 10^−3^ mg/mL)		Ti-Nano/P_HP-3
X	X		X (1 × 10^−5^ mg/mL)		Ti-Nano/P_HP-5

**Table 2 nanomaterials-12-02916-t002:** TaqMan primers used for real-time PCR reactions.

Gene	Gene Name	Identification
*Runx2*	Runt-related transcription factor 2 (*Runx2*)	Rn01512298_m1
*Alp*	Alkaline phosphatase (*Alp*)	Rn01516028_m1
*Ibsp*	Integrin binding sialoprotein (Bone sialoprotein, *Bsp*)	Rn00561414_m1
*Spp1*	Secreted phosphoprotein 1 (Osteopontin, *Opn*)	Rn00681031_m1
*Bglap*	Bone gamma-carboxyglutamic acid–containing protein (Osteocalcin, *Oc*)	Rn00566386_g1
*Col1a1*	Collagen type 1 alpha 1 (*Col*)	Rn01523366_m1
*Gapdh*	Glyceraldeyde-3-phosphaste dehydrogenase (*Gapdh*)	Rn01775763_g1

**Table 3 nanomaterials-12-02916-t003:** Contact angles of water on Ti-MP, Ti_Nano, Ti-Nano/HP, Ti-Nano/P_Ca, and Ti-Nano/P_HP1.

Sample	Contact Angle (°)	Standard Deviation
Ti-MP	70.0	2.5
Ti-Nano	31.4	1.4
Ti-Nano/HP	10.2	6.0
Ti-Nano/P_Ca	36.9	1.6
Ti-Nano/P_HP1	40.4	3.2

**Table 4 nanomaterials-12-02916-t004:** Content of total polyphenols on the samples (Folin–Ciocâlteu test).

Samples	GAE Units (mg/mL)	Standard Deviation
Ti-Nano	0	-
Ti-Nano/HPb	0	-
Ti-Nano/HP	0.0011	-
Ti-Nano/P_Ca	0.0020	0.0008
Ti_Nano/P_HP1	0.0082	0.0025

## Data Availability

Not applicable.

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
