# Peer review of "Functionalization with Polyphenols of a Nano-Textured Ti Surface through a High–Amino Acid Medium: A Chemical–Physical and Biological Characterization"

_nanomaterials, 2022, doi:10.3390/nano12172916_

Round 1
Reviewer 1 Report
The study by Scannavino et al. designed a nano-textured Ti surface functionalized with polyphenols through a high amino acid medium and explored its osteogenic effect in vitro. Through physisorption, functionalized PPHEs on Ti-Nano maintain their antioxidant activity. Osteoblastic cells interact in vitro with Ti-Nano functionalized with a solution of grape pomace extract resulting in the formation of mineralized matrix in all groups. This work provides some interesting results implying the potential uses of polyphenols-functionalized nano-textured Ti surface for fabricating osteogenic biomaterials. It is suggested for publication after the following concerns are addressed.
Specific comments:
1. For the better exhibition of the highlight of this work, it is suggested to add up the characterization of polyphenols and the reason why to choose it as a material to modify Ti surface in the introduction part.
2. The bars of the latter two pictures in Figure 3 are not very clear. Please refine them.
3. To illustrate the osteogenic ability of the modified Ti surfaces more evidently, please supplement the ALP staining and western blot results.
4. In 3.1.2 part, it is mentioned that “the aggregates clearly visible on the functionalized surfaces reveal to be almost completely composed by C (98%) when a spot EDS analysis is performed on them”. Please explain this phenomenon.
5. In the result of PCR, the Alp, Bsp, and Col mRNA expression levels increase in UMR-106 cells grown on Ti-Nano functionalized with 1 mg/mL extract (significantly for Alp and Col), whereas those of Oc and Opn reduced. Please discuss as to why this happened.
6. A number of statements are not supported by references. This manuscript may benefit from referring to more recent publications. Examples are as follows.
a) High-strength, porous additively manufactured implants with optimized mechanical osseointegration, Biomaterials 279 (2021) 121206.
b) Parathyroid hormone (1-34) can reverse the negative effect of valproic acid on the osseointegration of titanium rods in ovariectomized rats, J Orthop Translat 27 (2021) 67-76.
c) Promoting osteoblasts responses in vitro and improving osteointegration in vivo through bioactive coating of nanosilicon nitride on polyetheretherketone, J Orthop Translat 24 (2020) 198-208.
d) Three-dimensional biofabrication of an aragonite-enriched self-hardening bone graft substitute and assessment of its osteogenicity in vitro and in vivo, Biomaterials Translational 1(1) (2020) 69-81.
7. Please check the manuscript carefully and polish the language. For instance, in the line 355, “Ti-PM” should be changed to “Ti-MP”.
Author Response
Reviewer 1
The study by Scannavino et al. designed a nano-textured Ti surface functionalized with polyphenols through a high amino acid medium and explored its osteogenic effect in vitro. Through physisorption, functionalized PPHEs on Ti-Nano maintain their antioxidant activity. Osteoblastic cells interact in vitro with Ti-Nano functionalized with a solution of grape pomace extract resulting in the formation of mineralized matrix in all groups. This work provides some interesting results implying the potential uses of polyphenols-functionalized nano-textured Ti surface for fabricating osteogenic biomaterials. It is suggested for publication after the following concerns are addressed.
The authors thank the reviewer for the appreciation of the manuscript and the valuable comments.
Specific comments:
- For the better exhibition of the highlight of this work, it is suggested to add up the characterization of polyphenols and the reason why to choose it as a material to modify Ti surface in the introduction part.
A short paragraph with a description of the composition and potential benefits coming from the use of the extract selected for this research has been added to the introduction. We thank the reviewer for this comment that allowed us to improve the introduction of the paper.
- The bars of the latter two pictures in Figure 3 are not very clear. Please refine them.
Figure 3 has been corrected. We are sorry for the mistake.
- To illustrate the osteogenic ability of the modified Ti surfaces more evidently, please supplement the ALP staining and western blot results.
The UMR-106 cell line is composed of cells that are already committed to osteoblastic differentiation, as explained in this reference 1. We opted to detect bone sialoprotein (BSP) - a potent nucleator of apatite crystals that is associated with alkaline phosphatase in biomineralization foci - to compare the osteogenic potential of the functionalized surfaces and their control (no functionalization): you can find the basis of this approach discussed in this reference 2.
We selected two time-points: day 3 of culture is during its proliferative phase and day 7 is during its mineralization phase. Because of its restricted expression, BSP provides a valuable marker for osteoblast differentiation and bone formation, as described in 3.
Despite the unequivocal osteogenic potential of UMR-106 cells on all studied surfaces, which was also detected by the qualitative and quantitative analyses of Alizarin red staining (for the detection of calcium deposits), it was the PCR method that allowed to detect changes in the osteoblast markers expression during the cell culture growth on Ti-Nano functionalized with 1 mg/mL extract at day 5 of culture, when full osteoblast differentiation is achieved (please refer to answer 5 for a possible explanation for these results). We added this explanation to the manuscript for better clarification of the experimental setup and results.
1 C. M. Stanford, P. A. Jacobson, E. D. Eanes, L. A. Lembke, R. J. Midura., Rapidly forming apatitic mineral in an osteoblastic cell line (UMR 106-01 BSP). J Biol Chem. 270, 9420-8 (1995). doi.org/10.1074/jbc.270.16.9420
2 L. Malaval, N. M. Wade-Guéye, M. Boudiffa, J. Fei, R. Zirngibl, F. Chen, N. Laroche, J.P. Roux, B. Burt-Pichat, F. Duboeuf, G. Boivin, P. Jurdic, M.H. Lafage-Proust, J. Amédée, L. Vico, J. Rossant, J.E. Aubin. Bone sialoprotein plays a functional role in bone formation and osteoclastogenesis. J Exp Med. 205, 1145-53 (2008). doi.org/10.1084/jem.20071294
3 Y. Yang, Y. Huang, L. Zhang, C. Zhang. Transcriptional regulation of bone sialoprotein gene expression by Osx. Biochem Biophys Res Commun. 476, 574-579 (2016). doi.org/10.1016/j.bbrc.2016.05.164
- In 3.1.2 part, it is mentioned that “the aggregates clearly visible on the functionalized surfaces reveal to be almost completely composed by C (98%) when a spot EDS analysis is performed on them”. Please explain this phenomenon.
The sentence has been rewritten to be much clearer. The analysis EDS has been performed on an area to get the average chemical (elemental) composition of the surfaces and on some spots focused on the aggregates to get their specific composition.
- In the result of PCR, the Alp, Bsp, and Col mRNA expression levels increase in UMR-106 cells grown on Ti-Nano functionalized with 1 mg/mL extract (significantly for Alp and Col), whereas those of Oc and Opn Please discuss as to why this happened.
Despite the inherent complexity in studying and understanding the temporal expression of the key osteoblast markers at both tissue and single-cell levels (as discussed in 1 by the authors), the multiple comparisons among surface treatments on day 5 of culture – during the onset of matrix mineralization phase – indicate that UMR-106 cells grown on Ti-Nano functionalized with 1 mg/mL extract are likely in a more advanced stage of differentiation compared with other groups.
Indeed, the expression of Alp, Bsp, and Col mRNA is crucial for bone matrix production and mineralization. These are downstream genes that are regulated by Runx2 (see reference 2), which in turn has its expression modulated by polyphenols (see reference 3).
The observed downregulation of OPN, which is expected to occur when osteoblasts achieve full differentiation, also supports this interpretation. As last, an upregulation of OPN and OC – considered late osteoblast markers - would be expected to occur at later time points (not determined), during the mineralization phase of cultures (see references 1,2).
This discussion has been also added to the manuscript for clarification. We thank the referee because he/she allowed us to improve the discussion of our biological data.
1 P. Tambasco de Oliveira, S. Francis Zalzal, K. Irie, A. Nanci, Early Expression of Bone Matrix Proteins in Osteogenic Cell Cultures, J. Histochem. Cytochem. 51, 633–641 (2003), doi.org/10.1177/002215540305100509
2 T. M. Liu, E. H. Lee. Transcriptional regulatory cascades in Runx2-dependent bone development. Tissue Eng Part B Rev. 19, 254-63 (2013). doi.org/10.1089/ten.TEB.2012.0527
3 M.R. Byun, M.K. Sung, A.R. Kim, C.H. Lee, E.J. Jang, M.G. Jeong, M. Noh, E.S. Hwang, J.H. Hong. (-)-Epicatechin gallate (ECG) stimulates osteoblast differentiation via Runt-related transcription factor 2 (RUNX2) and transcriptional coactivator with PDZ-binding motif (TAZ)-mediated transcriptional activation. J Biol Chem. 289, 9926-35 (2014). doi.org/10.1074/jbc.M113.522870
- A number of statements are not supported by references. This manuscript may benefit from referring to more recent publications. Examples are as follows.
- a) High-strength, porous additively manufactured implants with optimized mechanical osseointegration, Biomaterials 279 (2021) 121206.
- b) Parathyroid hormone (1-34) can reverse the negative effect of valproic acid on the osseointegration of titanium rods in ovariectomized rats, J Orthop Translat 27 (2021) 67-76.
- c) Promoting osteoblasts responses in vitro and improving osteointegration in vivo through bioactive coating of nanosilicon nitride on polyetheretherketone, J Orthop Translat 24 (2020) 198-208.
- d) Three-dimensional biofabrication of an aragonite-enriched self-hardening bone graft substitute and assessment of its osteogenicity in vitro and in vivo, Biomaterials Translational 1(1) (2020) 69-81.
The authors thank the reviewer for the suggestion to improve the bibliography. Some papers here suggested or taken from the literature, as reported by the authors in answers number 3-5, have been added to the references of the manuscript.
- Please check the manuscript carefully and polish the language. For instance, in the line 355, “Ti-PM” should be changed to “Ti-MP”.
The language has been revised as requested.

Reviewer 2 Report
The authors propose a concept to convince the osteogenic activity by modifying the nanotextured Ti surface with polyphenols. Overall, the results not really justify the conclusions and the research design is unreasonable, such as the time point of mineralization.
Author Response
Reviewer 2
Comments and Suggestions for Authors
The authors propose a concept to convince the osteogenic activity by modifying the nanotextured Ti surface with polyphenols. Overall, the results not really justify the conclusions and the research design is unreasonable, such as the time point of mineralization.
We regret that the basis of our experimental approach was not clear as described in the manuscript. We here try to improve the explanation of it and then we will be glad to eventually discuss the details of our results with the referee, if necessary.
The approach used in the paper for the biological tests was not improvised, but it has been already described in a longstanding paper (see reference 1 published in 1995) with a huge number of citations (about 400 as reported in Scopus). Importantly, the mineralized matrix formation of UMR-106 cells under our culture conditions has been already previously described in detail by the authors, as you can see in reference 2.
Here we opted to use the UMR-106 cell line because these cells are in an advanced stage of osteoblastic differentiation and, thus, they are capable of producing a collagen matrix that undergoes mineralization in a shorter culture period: it means on days 5-7 of culture, that is 1-2 days after the confluence of the culture in our conditions. We also extended the mineralization analysis to day 10 of culture. The cell culture model we selected has also been applied to several in vitro studies on biomaterials by research groups worldwide during the last two decades (see references 3-12).
The description of the selected approach for the biological tests and discussion of the obtained results have been largely improved in the manuscript with respect to the first draft to be clearer. In any case, the authors are available to go further in a detailed discussion with the referee, if necessary for improving the paper.
1 C.M. Stanford, P.A. Jacobson, E.D. Eanes, L.A. Lembke, R.J. Midura, Rapidly Forming Apatitic Mineral in an Osteoblastic Cell Line (UMR 106—01 BSP), J. Biological Chemistry 270, 9420-9428 (1995), doi.org/10.1074/jbc.270.16.9420
2 W. M. A. Maximiano, E. Z. Marcelino da Silva, A. C. Santana, P. Tambasco de Oliveira, M. C. Jamur, C. Oliver, Mast Cell Mediators Inhibit Osteoblastic Differentiation and Extracellular Matrix Mineralization, J. Histochem Cytochem. 65, 723–741 (2017), doi: 10.1369/0022155417734174
3 Bosetti M, Lloyd AW, Santin M, Denyer SP, Cannas M. Effects of phosphatidylserine coatings on titanium on inflammatory cells and cell-induced mineralisation in vitro. Biomaterials. 26, 7572–7578 (2005). doi: 10.1016/j.biomaterials.2005.05.033
4 Variola F, Yi JH, Richert L, Wuest JD, Rosei F, Nanci A. Tailoring the surface properties of Ti6Al4V by controlled chemical oxidation. Biomaterials. 29, 1285–1298 (2008). doi: 10.1016/j.biomaterials.2007.11.040
5 Di Virgilio AL, Reigosa M, de Mele MF. Response of UMR 106 cells exposed to titanium oxide and aluminum oxide nanoparticles. J Biomed Mater Res A. 92, 80–86 (2010). doi: 10.1002/jbm.a.32339
6 Cortizo MC, Oberti TG, Cortizo MS, Cortizo AM, Fernández Lorenzo de Mele MA. Chlorhexidine delivery system from titanium/polybenzyl acrylate coating: evaluation of cytotoxicity and early bacterial adhesion. J Dent. 40, 329–337 (2012). doi: 10.1016/j.jdent.2012.01.008
7 Terzaki K, Kalloudi E, Mossou E, Mitchell EP, Forsyth VT, Rosseeva E, Simon P, Vamvakaki M, Chatzinikolaidou M, Mitraki A, Farsari M. Mineralized self-assembled peptides on 3D laser-made scaffolds: a new route toward 'scaffold on scaffold' hard tissue engineering. Biofabrication. 5, 045002 (2013). doi: 10.1088/1758-5082/5/4/045002
8 Prado Ferraz E, Pereira Freitas G, Camuri Crovace M, Peitl O, Dutra Zanotto E, de Oliveira PT, Mateus Beloti M, Luiz Rosa A. Bioactive-glass ceramic with two crystalline phases (BioS-2P) for bone tissue engineering. Biomed Mater. 9; 045018 (2017). doi: 10.1088/1748-605X/aa768e
9 Mooyen S, Charoenphandhu N, Teerapornpuntakit J, Thongbunchoo J, Suntornsaratoon P, Krishnamra N, Tang IM, Pon-On W. Physico-chemical and in vitro cellular properties of different calcium phosphate-bioactive glass composite chitosan-collagen (CaP@ChiCol) for bone scaffolds. J Biomed Mater Res B Appl Biomater. 105, 1758–1766 (2017). doi: 10.1002/jbm.b.33652
10 Augustine R, Dalvi YB, Yadu Nath VK, Varghese R, Raghuveeran V, Hasan A, Thomas S, Sandhyarani N. Yttrium oxide nanoparticle loaded scaffolds with enhanced cell adhesion and vascularization for tissue engineering applications. Mater Sci Eng C Mater Biol Appl. 103, 109801 (2019). doi: 10.1016/j.msec.2019.109801
11 Soubelet CG, Albano MP, Zuardi LR, Martorano AS, Castro-Raucci LMS, de Oliveira PT. Aging behavior of Y-TZP with bioglass addition and its impact on the flexural strength and osteoblastic cell response. Int J Appl Ceram Technol. 17, 2792–2806 (2020). doi.org/10.1111/ijac.13608
12 Jiang Y, Tao Y, Chen Y, Xue X, Ding G, Wang S, Liu G, Li M, Su J. Role of Phosphorus-Containing Molecules on the Formation of Nano-Sized Calcium Phosphate for Bone Therapy. Front Bioeng Biotechnol. 22, 875531 (2022). doi: 10.3389/fbioe.2022.875531
Round 2
Reviewer 2 Report
The quality of this manuscript has been improved partly after revision. However, the results not adequately support the conclusions. There are few major issues that may require the authors to address.
1. For the mRNA expression of osteogenic markers, the authors described “higher Alp, Bsp, and Col and lower Opn mRNA expression levels are detected in the cultures grown on Ti-Nano functionalized with the 1 mg/mL grape pomace extract solution”. But the difference of BSP expression was not significant among groups. Additionally, the expression of OCN significantly reduced, which is crucial for mineralization. If possible, please supplement more results by using WB.
2. The Alizarin red staining in Fig.8 wasn't obvious and the quantitative results also show no significant difference. Please provide high resolution pictures for revision.
3. There are many kinds of natural products with promoting cell osteogenic differentiation. If polyphenols only exhibit antioxidant activity in this study, please explain why polyphenols were selected.
4. In the disscussion, the authors mentioned “the extract displayed significant osteogenic effects ranging from 0.001 to 0.1 μg/mL in a pilot study”, whereas 1 mg/mL was selected in this work. If possible, it's highly suggested to measure the exact amount of organic components effectively adsorbed on samples.
5. In line 493 “the samples functionalized in the high- amino acid medium and Ti- Nano, as a control surface”, but the control group (Ti-Nano/HP) is missing in Figs. 5, 6, 7 and 8.
Author Response
Referee 2 – round 2
The quality of this manuscript has been improved partly after revision. However, the results not adequately support the conclusions. There are few major issues that may require the authors to address.
The authors thank the referee for appreciating the work they did and suggesting new improvements.
- For the mRNA expression of osteogenic markers, the authors described “higher Alp, Bsp, and Col and lower Opn mRNA expression levels are detected in the cultures grown on Ti-Nano functionalized with the 1 mg/mL grape pomace extract solution”. But the difference of BSP expression was not significant among groups. Additionally, the expression of OCN significantly reduced, which is crucial for mineralization. If possible, please supplement more results by using WB.
The reviewer has a point. In the Results, the description has been corrected, as follows: “The Alp, Bsp, and Col mRNA expression levels increase in UMR-106 cells grown on Ti-Nano functionalized with 1 mg/mL extract (significantly for Alp and Col), whereas those of Oc and Opn reduced (significantly for the latter).” Thus, the related sentences in the Discussion and in the Conclusions have been modified accordingly. As to the reviewer’s concern on OC expression, despite its key role in bone metabolism, during matrix mineralization, it acts as an inhibitor factor (refer to this review on the biology of osteocalcin 1). The manuscript has been improved and the reference 1 here reported has been added. An important study has been added to the references to mention that OPN has also an inhibitory effect on matrix mineralization 2. This concept reinforces our interpretation that, under our in vitro conditions, the surface functionalization with 1 mg/mL extract supports, at the molecular level, a differentiation osteoblast state compatible with the promotion of bone matrix formation. No WB analysis has been performed in the present study.
1 M. L. Zoch, T. L. Clemens, R. C. Riddle, New insights into the biology of osteocalcin, Bone 82, 42-49 (2016), doi.org/10.1016/j.bone.2015.05.046
2 W. N. Addison, D. L. Masica, J. J. Gray, M. D. McKee, Phosphorylation-dependent inhibition of mineralization by osteopontin ASARM peptides is regulated by PHEX cleavage, J Bone Miner Res. 25, 695-705 (2010), doi.org/10.1359/jbmr.090832
- The Alizarin red staining in Fig.8 wasn't obvious and the quantitative results also show no significant difference. Please provide high resolution pictures for revision.
As requested by the reviewer, a high-resolution figure of the stained UMR-106 cell cultures has been provided (attached in the pdf file). The qualitative results for the Alizarin Red staining were similar to the staining pattern observed for primary osteogenic cells, isolated by sequential trypsin/collagenase digestion of calvarial bone from newborn Wistar rats, and grown on a titanium surface structured at the micron and nanoscale (refer to the Figure 4 in reference 1 here reported). Even on a transparent, flatter surface (Thermanox coverslips), denser areas of matrix mineralization appear deep dark stained under transmitted light microscopy (please refer to the Figure 5B in reference 2 here reported); on a white material surface (Y-TZP + 64S bioglass), Alizarin Red-stained UMR-106 cultures exhibit a reddish appearance more evidently (please refer to Figure 11 in reference 3 here reported). No significant differences, in terms of Alizarin Red extraction/calcium content, could be due to the high osteogenic potential of the specific cell culture here used. In this context, the molecular effects observed on differentiated osteoblasts could likely result in a higher impact on bone matrix formation in a cell culture model with lower osteogenic potential, which might therefore be a subject for further investigation.
1 K. K.Y. Pereira, O. C. Alves, A. B. Novaes Jr et al., Progression of Osteogenic Cell Cultures Grown on Microtopographic Titanium Coated With Calcium Phosphate and Functionalized With a Type I Collagen-Derived Peptide, J. Periodontology 84, 1199-1210 (2013), doi.org/10.1902/jop.2012.120072
2 P. Tambasco de Oliveira, M. Andrade de Oliva, W. M. A. Maximiano et al., Effects of a Mixture of Growth Factors and Proteins on the Development of the Osteogenic Phenotype in Human Alveolar Bone Cell Cultures, J. Hystochemistry Cytochemistry 56, 629-638 (2008), doi.org/10.1369/jhc.2008.950758
3 M. F. Stábile, C. G. Soubelet, M. P. Albano et al., Effect of 64S bioglass addition on sintering kinetic, flexural strength and osteoblast cell response of yttria-partially stabilized zirconia ceramics, Int. J. Applied Ceramic Technology, 16, 517-530 (2019), doi.org/10.1111/ijac.13139
- There are many kinds of natural products with promoting cell osteogenic differentiation. If polyphenols only exhibit antioxidant activity in this study, please explain why polyphenols were selected.
There is a wide literature on the different benefits of polyphenols when tested or clinically used as a solution: anti-inflammatory properties, counteraction of the shift toward osteoclastogenesis in bone-loss pathologies, prevention of the destruction of the gingival connective tissue and alveolar bone in case of periodontitis of polymicrobial origin, estrogen and anticancer agents, reduction of arteries pressure (see references 1, 2, 3, 4, 5, 6 here reported). The idea at the basis of the research is that different biomedical devices can benefit from surface functionalization with polyphenols such as the cardiovascular, orthopaedic, and dental ones. In the case of bone contact applications, the process of bone repair might benefit from the paracrine secretion of several cell types: osteoblasts, macrophages, mast cells, and fibroblasts, whose functions are also modulated by polyphenols. A single research paper cannot cover all this work and it will be reported in other papers in the future. The introduction, last sentence of the discussion, and references have been improved to be much more effective and to make a wider scenario of applications.
1 M. Prakash, B.V. Basavaraj, K.N. Chidambara Murthy, Biological functions of epicatechin: Plant cell to human cell health, Journal of Functional Food 52 (2019) 14–24, doi.org/10.1016/j.jff.2018.10.021.
2 C. Braicu, M. R. Ladomery, V. S. Chedea, A. Irimie, I. Berindan-Neagoe, The relationship between the structure and biological actions of green tea catechins, Food Chem. 141 (2013) 3282–3289, doi.org/10.1016/j.foodchem.2013.05.122
3 A. Scalbert, I. T. Johnson, M. Saltmarsh, Polyphenols: antioxidants and beyond, Am. J. Clin. Nutr. 81 (2015) 215S–217S, doi.org/10.1093/ajcn/81.1.215S.
4 Torre E, Iviglia G, Cassinelli C, Morra M, Russo N. Polyphenols from grape pomace induce osteogenic differentiation in mesenchymal stem cells. Int J Mol Med. 45, 1721– 34 (2020).
5 Gómez- Florit M, Monjo M, Ramis JM. Identification of quercitrin as a potential therapeutic agent for periodontal applications. J. Periodontol.85, 966– 74 (2014)
6 Byun MR, Sung MK, Kim AR, Lee CH, Jang EJ, Jeong MG, et al., Epicatechin gallate (ECG) stimulates osteoblast differentiation via runt- related transcription factor 2 (RUNX2) and transcriptional coactivator
with PDZ- binding motif (TAZ)- mediated transcriptional activation. J Biol Chem.289, 9926– 35 (2014)
- In the discussion, the authors mentioned “the extract displayed significant osteogenic effects ranging from 0.001 to 0.1 μg/mL in a pilot study”, whereas 1 mg/mL was selected in this work. If possible, it’s highly suggested to measure the exact amount of organic components effectively adsorbed on samples.
The use of Gallic Equivalent Units, as obtained from the Folin & Ciocalteau test, is largely recognized as the main method to quantify the amount of total polyphenols in a solution. It quantifies the redox activity of the organic compounds by using the chemical activity of gallic acid, a simple model molecule, as a measure unit. This is because the redox chemical activity is the basis of the biological and chemical effects of the polyphenols and it is of great relevance. Despite of the number or mass of biomolecules per volume, it is their chemical reactivity that is of interest. The test has been adapted by the authors to be used on solid samples to quantify the amount of adsorbed biomolecules.
- In line 493 “the samples functionalized in the high- amino acid medium and Ti- Nano, as a control surface”, but the control group (Ti-Nano/HP) is missing in Figs. 5, 6, 7 and 8.
The use of the term “control sample/surface” has been checked in the whole manuscript and ambiguous sentences have been modified. Concerning the biological tests, see the following sentence (paragraph 3.2): “According to the chemical-physical characterization, the biological experiments are performed only on the samples functionalized in the high- amino acid medium, that is more promising, and Ti-Nano is used as a control surface for all biological tests”.
